# Therapeutic Approach to Alzheimer’s Disease: Current Treatments and New Perspectives

**DOI:** 10.3390/pharmaceutics14061117

**Published:** 2022-05-24

**Authors:** Teresa Pardo-Moreno, Anabel González-Acedo, Antonio Rivas-Domínguez, Victoria García-Morales, Francisco Jose García-Cozar, Juan Jose Ramos-Rodríguez, Lucía Melguizo-Rodríguez

**Affiliations:** 1Instituto Nacional de Gestión Sanitaria (INGESA), Primary Health Care, 51003 Ceuta, Spain; terepardo@correo.ugr.es; 2Biomedical Group (BIO277), Department of Nursing, Faculty of Health Sciences (Melilla), University of Granada, 52005 Granada, Spain; anabelglez@correo.ugr.es; 3Department of Celular Biology, University of Seville, 41009 Seville, Spain; ardominguez@us.es; 4Department of Biomedicine, Biotechnology and Public Health, Physiology Area, Faculty of Medicine, University of Cádiz, Pl. Falla, 9, 11003 Cádiz, Spain; 5Department of Biomedicine, Biotechnology and Public Health, Immunology Area, University of Cádiz Pl. Falla, 9, 11003 Cádiz, Spain; curro.garcia@gm.uca.es; 6Institute of Biomedical Research Cádiz (INIBICA), Hospital Universitario Puerto del Mar, Novena Planta Investigación Avda. Ana de Viya, 21, 11009 Cádiz, Spain; 7Department of Physiology, Faculty of Health Sciences (Ceuta), University of Granada, 51001 Ceuta, Spain; juanjoseramos@go.ugr.es; 8Biomedical Group (BIO277), Department of Nursing, Faculty of Health Sciences, University of Granada, 18016 Granada, Spain; luciamr@ugr.es; 9Instituto de Investigación Biosanitaria, Ibs Granada, Av. de Madrid, 15, 18012 Granada, Spain

**Keywords:** Alzheimer’s disease, dementia, Aβ pathology, tau pathology, pharmacology, drugs

## Abstract

Alzheimer’s disease (AD) is the most common cause of dementia. The pathophysiology of this disease is characterized by the accumulation of amyloid-β, leading to the formation of senile plaques, and by the intracellular presence of neurofibrillary tangles based on hyperphosphorylated tau protein. In the therapeutic approach to AD, we can identify three important fronts: the approved drugs currently available for the treatment of the disease, which include aducanumab, donepezil, galantamine, rivastigmine, memantine, and a combination of memantine and donepezil; therapies under investigation that work mainly on Aβ pathology and tau pathology, and which include γ-secretase inhibitors, β-secretase inhibitors, α-secretase modulators, aggregation inhibitors, metal interfering drugs, drugs that enhance Aβ clearance, inhibitors of tau protein hyperphosphorylation, tau protein aggregation inhibitors, and drugs that promote the clearance of tau, and finally, other alternative therapies designed to improve lifestyle, thus contributing to the prevention of the disease. Therefore, the aim of this review was to analyze and describe current treatments and possible future alternatives in the therapeutic approach to AD.

## 1. Introduction

### 1.1. Epidemiology of Alzheimer’s Disease

Currently, approximately 35.6 million people worldwide suffer from dementia, and 7.7 million new cases are diagnosed each year. Recent studies have confirmed this trend, predicting an 87% increase in Europe between 2010 and 2050 [1]. Alzheimer’s disease (AD), which is the most common cause of dementia, accounts for 60–75% of all cases [2] Moreover, while deaths caused by other health problems, such as heart disease, have decreased in recent years, those attributed to AD have increased by 68% in the last decade alone [3]. The main risk factor in the etiopathogenesis of AD is age, and as life expectancy progressively increases, so does the number of people affected. Studies that have evaluated AD show that the annual prevalence in people aged 45–64 years is approximately 24.2/100,000, and the incidence is 6.3/100,000 [4]. However, the disease is more common in people over 65 years of age, and the likelihood of developing AD increases exponentially with age, doubling every 5 years thereafter.

### 1.2. Physiopathology of Alzheimer’s Disease

AD is characterized by a predominant impairment of episodic memory. This symptom is often accompanied by a multitude of cognitive impairments in areas such as executive function, language, visuospatial ability, and decision making [2]. Thus, AD appears as a progressive deterioration of higher brain functions, also affecting decision-making ability. Alzheimer’s patients survive an average of 7–10 years after diagnosis [3]. Moreover, this disease currently has no unequivocal premortem diagnosis, and can only be diagnosed histologically postmortem by the presence of senile plaques (SP), neurofibrillary tangles, and neuronal and synaptic loss [5], which makes early identification and treatment even more difficult.

#### 1.2.1. Amyloid-β Pathology

SP are a classic neuropathological feature in AD-affected brains and remain a feasible origin of synaptic and neuronal loss. SP are the result of a progressive accumulation of parenchymal amyloid-β (Aβ) [5]. Aβ is a 39–43 amino acid peptide derived from the progressive processing of Aβ precursor protein (APP) by β- and γ-secretase complexes following an amyloidogenic pathway. The amyloidogenic pathway is enhanced in AD, and mutations in the APP peptide or in the β-secretase enzyme may promote or accelerate the onset of the pathology. While APP and β-secretase mutations are responsible for most known cases of AD, little is yet known about the causes of dementia, as the vast majority of cases (~95%) are sporadic, and these patients accumulate Aβ without known mutations. Therefore, it has been proposed that alterations in Aβ degradation or clearance may also play a key role in the pathogenesis of AD [6]. The toxic effects of Aβ described, both in humans and in experimental models, range from free oligomers to compact SP, and different Aβ species have been associated with synaptic loss and the development of neuritic dystrophies [7]. Furthermore, some studies have suggested that the accumulation of Aβ, like SP, may also contribute to the loss of dendritic spines [8]. In addition, senile compact plaques have also been associated with the abnormal curvature of nearby neurites and may alter cortical synaptic integration [7]. It has been suggested that Aβ per se can even promote neuronal death in the hippocampus and entorhinal cortex in AD development [9]. Moreover, these areas are particularly relevant in learning and memory, and therefore, are very vulnerable in this disease. Aβ deposition also occurs at the vascular level as cerebral amyloid angiopathy (CAA), which is present in most AD patients, causing damage or impairment to the blood-brain barrier (BBB), affecting its functionality [10]. However, CAA can also occur in the absence of AD [11], remaining a possible exponent of vascular dementia (VaD).

#### 1.2.2. Tau Pathology

Aβ pathology precedes the other major neuropathological feature of AD, the hyperphosphorylation and aggregation of tau protein into neurofibrillary tangles. Tau is a microtubule-associated protein that is abundantly expressed in the brain, binding to tubulin to promote microtubule assembly. It supports other cytoskeletal structures and regulates several major functions in the neurons [12]. Tau protein plays an important role in the pathogenesis of AD, as when it is abnormally phosphorylated, it can be found as an intraneuronal deposit, forming filamentous aggregates in the soma and proximal dendrites [13]. Neurofibrillary tangles constitute masses of argyrophilic fibers that can stain intensely with thioflavin-S. It appears that tau protein deposition in Alzheimer’s patients increases proportionally with the duration and severity of the disease. Furthermore, it is generally accepted that tau dysfunction is one of the main proximal causes of neuronal loss in AD, although neurofibrillary tangles appear to be downstream pathological processes [5].

#### 1.2.3. Neuroinflammation and Neuronal Loss

Neuronal loss is the histological feature that best correlates with the severity and duration of dementia [5]. It is also the main cause of cortical atrophy observed in AD. The distribution of neuronal loss correlates with the location of neurofibrillary tangles, although the presence of neurofibrillary tangles is not sufficient to explain the enormous neuronal loss observed in AD patients [14]. In addition to SP and tau hyperphosphorylation, neuroinflammation and oxidative stress also play an important role in neurodegeneration. Inflammation is a two-sided mechanism. On the one hand, it promotes the secretion of proinflammatory factors, such as cytokines, which lead to increased cerebral blood flow to the affected area and the removal of damaged tissue by microglial cells [15,16]. In this regard, microglial cells play a key role in the first line of immune defense. They are involved in phagocytic processes and play a crucial role in the attempt to eliminate toxic products and release cytotoxic factors, and can also act as antigen-presenting cells. On the other hand, an excessive inflammatory response can cause tissue damage, promote chronic inflammation, and eventually lead to neuronal death [5]. In addition, the increased production of reactive oxygen species can damage the BBB. Oxidative stress plays a detrimental role in the pathogenesis of neuronal death in AD by damaging different cellular elements, such as proteins, lipids, or nucleic acids, which accumulate as oxidized/damaged macromolecules and are not efficiently removed and renewed by antioxidant enzymes [17]. Following this idea, a reduction in, or the loss of function of antioxidant enzymes in AD has been reported as a possible contributor to neuronal death [18].

## 2. Results

### 2.1. Approved Treatment for Alzheimer’s Disease

Despite the increase in the number of cases in recent years and the associated socio-economic costs, there is still no effective treatment to reverse or slow the progression of AD. So far, only six drugs have been approved by the US Food and Drug Administration (FDA): aducanumab, donepezil, galantamine, rivastigmine, memantine, and a manufactured combination of memantine and donepezil (Figure 1). Of these, aducanumab is the only one used for the clearing of Aβ plaques, while the remaining five are focused on symptomatological treatment acting at two levels: through agonism of the cholinergic system or as antagonists of the *N*-methyl-D-aspartate receptor (NMDA-receptor).

#### 2.1.1. Aducanumab

Aducanumab is a monoclonal antibody specific for soluble b-amyloid protein fibrils and oligomers whose application is aimed at the clearance of Aβ plaques. The recommended dose of aducanumab is 10 mg/kg, administered by IV over one hour every 4 weeks, and at least 21 days apart [19]. This drug was recently approved in April 2021 by the FDA, as it is considered to be able to act on SP, slowing the progression of AD. This approval was based on the results of three clinical trials that provide strong evidence of the effectiveness of this treatment. The first, PRIME (NCT01677572), a multicenter, randomized, 12-month, double-blind, placebo-controlled, multiple-dose study, enrolled 165 individuals who were administered monthly intravenous (IV) pull doses of aducanumab (1, 3, 6, or 10 mg kg^−1^) for one year. This study showed a dose-time-dependent reduction in Aβ plaques and an improvement in clinical manifestations of AD, as assessed by the Clinical Dementia Rating-Sum of Boxes (CDR-SB) and Mini Mental State Examination (MMSE) scores for the highest dose [20]. The other two studies, EMERGE (NCT02484547) and ENGAGE (NCT02477800), 2 double-blind, placebo-controlled, parallel-group, phase 3 randomized clinical trials with 1643 and 1653 participants respectively, showed contradictory results. While the EMERGE study showed a 22% decrease in CDR-SB scores, and a significant improvement in questionnaires such as the MMSE, the Alzheimer’s Disease Assessment Scale-Cognitive Subscale (13 Items) (ADAS-Cog 13), and the Alzheimer’s Disease Cooperative Study-Activities of Daily Living Inventory (Mild Cognitive Impairment Version) (ADCS-ADL-MCI) after administration of Aducanumab, 10 mg/kg IV monthly, the ENGAGED study did not reproduce these results, probably due to differences between the studies, in the course of the disease, and in the response according to the number of treatments received. However, additional pharmacometric analyses showed an association between aducanumab administration and response to the aforementioned questionnaires observed in the two phase 3 studies. These studies also showed, in a subgroup analysis of participants, significant reductions in amyloid deposition in both subgroup studies for high-dose aducanumab treatments (10 mg/kg) versus placebo, but significant reductions in cerebrospinal fluid (CSF) tau proteins were only observed for the EMERGE study [21,22]. On the other hand, the main adverse effect was amyloid-related imaging abnormalities—manifesting as edema or hemosiderin deposition—which was reported in 41% of patients versus 10% of those taking the placebo. Most cases were symptom-free, and adverse effects resolved in most patients [23].

However, the European Medicines Agency has withdrawn the marketing authorization for this product for the treatment of Alzheimer’s disease due to interactions with the CHMP, indicating that the data provided thus far would not be sufficient to support a positive opinion on the effectiveness of the product. Thus, following FDA amendment, this treatment is limited to cases of mild Alzheimer’s disease or mild cognitive impairment due to Alzheimer’s disease, as reported in the trials. In this case, patients must have a brain MRI before starting treatment. These MRI scans should be repeated at 7 and 12 months to assess the presence of brain edema, microhemorrhages, and superficial siderosis [19].

#### 2.1.2. Acetylcholinesterase Inhibitors (AChE)

Tacrine

Tacrine was the first drug in this group to be approved by the FDA in September 1993. It is an acridine-derived cholinergic that works as a central inhibitor of acetylcholinesterase, increasing acetylcholine levels in various brain regions [24]. Its action in inhibiting pseudocholinesterase is more potent than acetylcholinesterase itself. Tacrine was first tested in AD by Dr. Williams Summer in 1989. Its main advantages are its ability to be administered orally and venously and its ability to cross the BBB. However, as its use became more widespread, its effectiveness began to be questioned due to conflicting results in clinical trials [25,26,27,28]. Thus, it was concluded that tacrine had a palliative effect in the treatment of patients with mild to moderate dementia, but did not alter the course of the underlying neurodegeneration [29]. Its use was discontinued in 2013 due to a large number of adverse effects such as nausea, vomiting, loss of appetite, diarrhea, and clumsiness, as well as hepatic cytotoxicity that has been reported as a consequence of its administration [30,31].

2.Donepezil

Donepezil is a piperidine-derived cholinergic drug, a central reversible, non-competitive inhibitor of acetylcholinesterase, the enzyme responsible for the hydrolysis of acetylcholine. Moreover, donepezil also acts at the molecular and cellular level in the pathogenesis of AD, causing inhibition of various aspects of glutamate-induced excitotoxicity, reduction of early expression of inflammatory cytokines, induction of a neuroprotective isoform of AChE, and reduction of oxidative stress-induced effects. [32]. It was approved in 1996 for the therapeutic management of mild to moderate cases of AD [33]. Donepezil can be administered orally in tablet, liquid, or jelly form, or transdermally. In mild to moderate dementia, it is recommended to start treatment with 5 mg/day, increasing to 10 mg/day for four to six weeks. For patients with moderate to severe dementia, the dose could be increased to 23 mg/day, as long as the patient has been on the 10 mg/day dose for at least 3 months [33]. Donepezil administration at a dose of 10 mg/day has been shown to improve cognitive function, basic activities of daily living, and clinician-rated global impression scores, with no improvement in behavior or quality of life [34,35,36,37]. In addition, the effectiveness of doses up to 23 mg/day has been studied, and no significant differences have been found compared to a dose of 10 mg/day [38,39]. However, none of the doses studied has been able to interrupt the progression of AD [40]. On the other hand, this drug is characterized by good patient tolerance and mild adverse effects, especially on the gastrointestinal tract or the nervous system [41].

3.Galantamine

Galantamine is a selective tertiary isoquinoline alkaloid that acts as a selective, competitive, and reversible acetylcholinesterase inhibitor. In addition, it stimulates the intrinsic action of acetylcholine on nicotinic receptors [42]. This drug was approved by the FDA in 2001 [43]. The route of administration is oral, with doses of 4, 8, 12, 16, and 24 mg, either as a quick-release solution (twice a day), or as extended-release capsules (once a day). The initial dose proposed for treatment is 8 mg/day with an increase, as a maintenance dose, up to 16 mg/day twice a day after 4–8 weeks [44]. In terms of its application in AD, what is really interesting about galantamine is its ability to act at the level of the central nervous system, with little activity in the peripheral system. In this line, galantamine has been associated with different molecules that favor its delivery in the brain, such as ceria-containing hydroxyapatite particles, solid lipid nanoparticles, or chitosan [45,46,47]. The clinical trials developed on this pathology have shown that this treatment is able to reduce the behavioral and cognitive symptoms (agitation, anxiety, disinhibition, and aberrant movements) in patients with mild to moderate AD [44,48,49]. A recent meta-analysis conducted by Li et al. showed that this drug is not only effective in controlling behavioral symptoms, but it also improves the performance of basic activities of daily living, cognitive function, and clinicians’ perceptions of global state, which makes it the drug of choice for the treatment of AD [50]. Although this drug has shown good safety and tolerability, its use is not without adverse effects, such as convulsions, severe nausea, stomach cramps, vomiting, irregular breathing, confusion, muscle weakness, and watering eyes [51].

4.Rivastigmine

This pharmaceutical agent was introduced in Switzerland in 1997 and approved by the FDA in 2000; is indicated for mild and moderate AD, as well as mild-moderate Parkinson’s dementia. It is a pseudo-irreversible inhibitor of AChE and butyrylcholinesterase that acts by binding to the anionic and stearic site of AChE [52]. It is available as tablets or drops, administered at an initial dose of 1.5 mg/12 h, which can be increased according to tolerability up to 6 mg/12 h, at intervals of at least 2 weeks from the previous dose, the most effective dose being between 3–6 mg/12 h. This drug is also available in transdermal patches, the initial dose of which is 4.6 mg/day, although, if tolerability is adequate, the dosage can be increased to 13.3 mg/day [53]. This route also allows continuous release for 24 h, avoiding gastrointestinal effects due to intestinal and hepatic metabolism. Similarly, transdermal patches are a particularly interesting option in AD patients, who often have memory loss and impaired swallowing, making it difficult to use the oral route [54]. In addition, cost-effectiveness studies have shown that this method of delivery is the optimal treatment strategy in economic terms [55]. Trials that have evaluated the use of rivastigmine in the management of AD found an improvement in cognitive function as assessed by the ADAS-Cog and MMSE, activities of daily living, as well as an improvement in clinician-rated global impression of change after 26 weeks of treatment, using doses of 6–12 mg daily orally, or 9.5 mg daily transdermally [56,57,58,59,60,61]. However, it should be noted that patients taking rivastigmine are not very adherent to treatment due to adverse effects, including stomach pain, weight loss, diarrhea, loss of appetite, nausea, and vomiting. Moreover, an overdose of this drug may cause numerous symptoms, including irregular, (fast or slow) breathing, chest pain, and slow or irregular heartbeat [62].

#### 2.1.3. N-Methyl D-Aspartate (NMDA) Antagonists

Memantine

Memantine is a non-competitive, moderate affinity, voltage-dependent NMDA receptor antagonist. It blocks the effects of pathologically elevated glutamate tonic levels that can lead to neuronal dysfunction. The malfunction of glutamate-mediated neurotransmission, particularly at the NMDA receptors, contributes to both symptom expression and progression of AD to neurodegenerative dementia [63]. In 2003, memantine became the first drug approved by the US FDA to treat moderate-to-severe AD [64]. It is available as slow-release capsules of 7, 14, 21, and 28 mg; as a 2 mg/mL solution, and in 10 mg tablets, with an initial dose of 5 mg/day for the first week. Thereafter, the dose is increased to 5 mg twice a day during the second week, and in the third week, 15 mg is administered in the form of one 10 mg tablet in the morning and half of another tablet in the afternoon. From the fourth week on, treatment can be continued at the recommended maintenance dose of 20 mg a day (one tablet twice a day) [63,65]. The administration of memantine in patients with moderate to severe AD showed a small improvement in global clinical rating: 0.21 points on the CIBIC-Plus, in cognitive functioning: 3.11 points on the Severe Impairment Battery (SIB), in performance in activities of daily living: 1.09 points on the ADCS-ADL scale, and in behaviour and mood: 1.84 points on the Neuropsychiatric Inventory [66]. However, according to recent meta-analyses, memantine does not seem to be equally effective in patients with mild AD, as no significant differences were found with respect to placebo in the previously mentioned parameters [50,66]. In terms of adverse effects, the following should be noted: dizziness, headache, confusion, diarrhea, and constipation, although fatigue, pain, hypertension, weight gain, hallucination, confusion, aggressive behavior, vomiting, abdominal pain, and urinary incontinence may also appear less frequently [67].

On the other hand, the combination of memantine with donepezil (Namzaric^®^) for the symptomatic treatment of moderate to severe Alzheimer’s disease is also approved by the FDA. However, another drug based on memantine hydrochloride and donepezil hydrochloride, known commercially as Acrescent^®^, has not been approved by the European Medicines Agency, due to a lack of consistent evidence of its effectiveness in this pathology [68,69]. This combination of drugs prevents the toxic effects associated with excess glutamate, as well as the breakdown of acetylcholine in the brain. According to scientific evidence, Memantine plus donepezil is more effective in improving cognition, as assessed by the ADAS-Cog and SIB scale, global assessment, daily activities, and neuropsychiatric symptoms compared with placebo, but has lower acceptability than monotherapy or placebo [70]. Regarding the cost-effectiveness of this treatment, the scientific literature does not show completely clear results. While authors such as Cappel et al. or Weycker et al. suggest that combination therapy is more cost-effective than donepezil alone, Knapp et al. found no significant differences between the two treatments [71,72,73].

### 2.2. Pharmacological Treatments under Investigation

Due to the absence of, and the high demand for successful treatments to prevent and slow the progression of AD, research in recent decades has focused on the typical pathophysiological mechanisms of the disease [74] (Figure 2). In relation to Aβ pathology, the main pharmacological treatments under investigation can be classified into those that decrease Aβ42 production, those that reduce Aβ load in SP, and those that enhance Aβ clearance (Table 1). In the case of tau pathology, treatments under study include inhibitors of tau protein hyperphosphorylation and aggregation, as well as those that potentiate tau protein clearance [75] (Table 2).

#### 2.2.1. Aβ Pathology

γ-secretase inhibitors

One of the therapeutic targets in the treatment of AD is the inhibition of γ-secretase, which acts on the APP through its sequential cleavage. However, this enzyme also works at other levels, cleaving transmembrane proteins such as the Notch 1 receptor, an important component in cell communication and differentiation. [76]. This could explain the failure of treatments based on this pathway in various clinical trials to date. For example, a phase 3 clinical trial showed that Semagacestat (LY-450139) did not improve cognitive abilities and, in the case of patients receiving a higher dose, caused a worsening of functional capacity, as well as an increased risk of skin cancer and infections [77]. In the case of Avagacestat (BMS-708163), a greater progression of disease symptoms and brain atrophy was also observed compared to the control group, as well as numerous side effects including skin cancer and renal dysfunction [78]. Another therapeutic alternative included in this group is Tarenflurbil, a γ-secretase inhibitor administered intranasally. However, the phase 3 clinical trial based on this drug was stopped due to poor brain penetration [79]. Due to the side effects of γ-secretase inhibitors, their role in the pharmacological approach to AD is currently being challenged [80].

2.β-secretase inhibitors

β-secretase inhibitors aim to lower Aβ levels in CSF, and include drugs such as Lanabecestat, Verubecestat, Atabecestat, Elenbecestat and Umibecestat. [81]. However, currently only two of these are still under study: Elenbecestat (E2609) and Umibecestat (CNP520) in phase 2 and 3, respectively [80]. Subjects included in these trials were individuals at high risk for AD: age in the range of 60–75 years, APOE4 genotype and heterozygotes (APOE ε2/ε4 or ε3/ε4), and high levels of amyloid in the brain [82]. The remaining treatments were discontinued due to lack of efficacy or adverse effects associated with their use, such as rash, liver toxicity, and neuropsychiatric symptoms [83,84,85]. Despite these incidents, it is noteworthy that all treatments significantly reduced CSF Aβ levels, although this did not result in cognitive or functional improvement [86].

3.α-secretase modulators

The non-amyloidogenic pathway is triggered after the cleavage of APP by α-secretase, so another pharmacological target could be the modulation of this enzyme. According to the scientific literature, the activation of α-secretase is mediated through the phosphatidylinositol 3-kinase (PI3K)/Akt pathway via γ-aminobutyric acid (GABA) receptor signaling. Thus, drugs that can activate this pathway, or whose action resembles that of selective GABA receptor modulators, could constitute a new therapeutic approach in AD [75]. Within this group is Etazolate (EHT0202), a GABA receptor modulator that has already been tested in a phase 2 clinical trial demonstrating its safety in patients with mild to moderate AD. However, a phase 3 trial is still pending [87]. On the other hand, APH-1105 and ID1201 are drugs that activate the PI3K/Akt pathway and are currently being studied in phase 2 clinical trials. APH-1105 is an intranasal drug whose safety, tolerability, and efficacy are being evaluated for the treatment of subjects with mild-moderate AD. ID1201 is a Melia toosendan fruit extract for use in subjects with mild to moderate AD [80].

4.Aggregation inhibitors

This group of drugs aims to prevent the formation of Aβ42 fibers characteristic of Aβ pathology. Scyllo-inositol (ELND005) was the last aggregation inhibitor tested in humans in a phase 2 clinical trial. It was discontinued due to limited evidence to support its benefit, as well as dose-dependent toxicity from its use. [88]. Currently, research is directed towards the use of peptidomimetics, which inhibit or reverse the aggregation of Aβ42. These include KLVFF, whose peptide sequence is similar to the hydrophobic core portion of Aβ and gradually replaces it. However, its action is not limited to preventing aggregation, but can also dissolve oligomers that are resistant to proteolytic breakdown [89]. A new class of peptidomimetics under investigation are the γ-AApeptides, which may be up to 100 times more efficient than KLVFF as aggregation inhibitors. The latter have yet to be tested in in vivo trials [90].

5.Metal interfering drugs

Another etiological factor related to the development of AD is dyshomeostasis, or abnormal accumulation of metal ions such as copper, iron, or zinc [76]. In this regard, deferiprone acts as a chelating agent that helps to decrease iron stores. This treatment is currently being studied in a phase 2 clinical trial of subjects with mild and prodromal AD [91]. Another drug in this group is PBT2, which showed promising results in preclinical trials and is currently in a phase 2 clinical trial. This drug has demonstrated the ability to decrease Aβ in CSF by 13%, as well as a dose-dependent improvement in executive functions in subjects with early AD [92,93].

6.Drugs that enhance Aβ clearance (immunotherapy)

This group of drugs belongs to the immunotherapeutic approach and includes two possible modalities: active and passive immunotherapy [94]. Active immunotherapy involves the use of whole proteins or protein fragments that promote the creation of antibodies by B cells, thereby developing the patient’s immune response [86]. Passive immunotherapy involves the passive inoculation of monoclonal antibodies (mAbs) or polyclonal antibodies that act against Aβ peptides, making the inflammatory process developed by T cells unnecessary [95].

With regard to active immunotherapy, six drugs are currently under study in phase 1, 2 and 3 clinical trials. On the one hand, CAD106 and CNP520 have been studied in two simultaneous phase 3 trials in subjects at risk for AD. However, in 2020 these studies were stopped due to the detection of changes in cognitive function, brain volume loss, and body weight [82]. In addition, vaccines have been developed to create anti-Aβ40 antibodies, including ABvac40, GV1001, ACC-001, UB-311, and AF20513. These vaccines, with the exception of AF20513, are still in phase 2 clinical trials and have obtained generally promising results. Thus, treatment with ABvac40 resulted in the development of Aβ40 antibodies in 92% of patients administered the drug [96]. In the case of ACC-001, this was administered in subjects with mild to moderate AD, with higher Ig G and anti-Aβ40 levels than in the other groups [97]. UB-311 induced an immune response in 100% of the sample, as well as a slowing of cognitive decline. In addition, only mild adverse effects, such as swelling at the puncture site and agitation, were observed [98].

As for passive immunotherapy based on mAbs, we can currently identify several drugs in development. On the one hand, crenezumab has been associated with an improvement in ADAS-Cog scores in individuals with low advanced AD, being sufficiently safe and tolerable [99]. On the other hand, treatment with gantenerumab has achieved a progressive decrease in Aβ in recent clinical trials. A phase 2 study involving individuals with prodromal AD, and a phase 3 clinical trial studying the efficacy of this treatment at disease onset are currently underway [100,101]. Finally, one of the most promising drugs is LY3002813, whose use was associated with an improvement in ADAS-Cog scores and a decrease in Aβ to half its initial levels at a dose of 10 mg/kg [102].

It is important to mention that, within this type of immunotherapy, there are fields of research aimed at obtaining human anti-Aβ from intravenous immunoglobulin (IVIg). IVIg is made up of monoclonal antibodies, mostly IgG, and is obtained through plasma donations from healthy individuals. In one phase 3 clinical trial, participants with mild and moderate AD showed good tolerability, but no decrease in cognitive and functional impairment [75]. Another treatment under investigation is plasma exchange via albumin. The aim of this treatment is to eliminate plasma Aβ by removing it and replacing it with albumin. In this way, albumin promotes the transport of Aβ present in CSF into the plasma to repair the imbalance between plasma and brain Aβ levels, thus allowing its elimination [103].

#### 2.2.2. Tau Pathology

Inhibitors of tau protein hyperphosphorylation

There are several types of kinases that promote tau phosphorylation, so their inhibition has been proposed as a possible pharmacological approach in AD. In this sense, one of the pharmacological targets to prevent tau protein hyperphosphorylation is glycogen synthase kinase 3 beta (GSK3β). One of the most studied GSK3β inhibitors today is lithium chloride, a drug widely used for affective disorders, which has been shown to produce a decrease in tau hyperphosphorylation and aggregation in transgenic mice. These observations prompted the development of a phase 2 clinical trial in AD patients. However, despite the good results obtained in in vivo studies, no improvement in cognitive status, and no reduction of tau in the CSF of patients under treatment was observed [104,105]. Another drug that has obtained positive results is tideglusib, which has been tested in a transgenic mouse model of AD, showing reduced tau phosphorylation and reduced Aβ peptide accumulation, preventing neuronal death and maintaining cognitive function. Despite this, its use in a phase 2 clinical trial involving individuals with mild to moderate AD failed to reduce cognitive impairment [106].

2.Tau protein aggregation inhibitors

Methylene blue has been proposed as an inhibitor of tau protein aggregation. Thus, some authors have described an increase in cognitive abilities in an in vivo model with transgenic mice, as well as an improvement in the ADAS-Cog score, after administering this compound to a group of subjects suffering from AD. Subsequently, a derivative called TRx0237 (LMTM) was studied, which had better absorption and tolerability characteristics. However, in two phase 3 clinical trials involving patients with mild to moderate AD, no decrease in cognitive impairment was observed compared to the control group [107].

3.Drugs that promote the clearance of tau (immunotherapy)

The results obtained in different in vivo assays suggest that phosphorylated tau (p-tau) epitopes could be a therapeutic alternative to induce an immune response that increases tau protein clearance. This group includes both active and passive immunotherapy [108]. Active immunotherapy focuses on inducing an autoimmune response against tau pathology and is used in mild to moderate AD. This group includes AADvac-1, which is the first anti-tau vaccine administered in humans. This vaccine obtained favorable and safe results, including the production of high affinity anti-tau antibodies in most patients, reduction of blood neurofilament and p-tau in CSF, and a slowing of cognitive decline in younger patients [109]. Another therapeutic alternative included in this group is ACI-35. However, the immune response triggered was lower than expected, so this drug was modified to create ACI-35.030, which has obtained good results in terms of safety and immunity with minimal doses in phase 1b/2a trials [107].

Passive immunization includes C2N-8E12 (tilayonemab), which acts directly in the extracellular space. This treatment has been associated with the disruption of p-tau deposition and improved cognitive function after administration in transgenic mouse models in a phase 1 clinical trial [104]. A trial has also been conducted with bepranemab (UCB0107), a human recombinant, full-length IgG4 monoclonal antibody aimed at testing the safety, tolerability, and pharmacokinetics of this monoclonal antibody, although the results have not yet been published [110,111]. In addition to those mentioned above, there are other anti-tau mAbs under investigation, such as BII076, JNJ-63733657, or LY3303560 [80].

#### 2.2.3. Other Treatments under Investigation

##### Nanomedicine Strategies

Different nanobiomedicine treatments have been tested to facilitate targeted delivery to the specific part of the AD-affected brain, while preserving the healthy brain, but also to overcome obstacles associated with drug delivery, such as BBB penetration (Table 3). Nanocarriers have been made to increase the specificity of the drug to its target and are being tested to facilitate drug entry into the brain [112]. The blood-brain barrier has numerous impediments that limit drug entry, so nanotransporters, lipotransporters, carbon nanotubes, and others have been tested to improve the efficacy and bioavailability of the drug in the central nervous system [113]. One solution to this problem is based on the design of nanoparticles with high affinity for the endothelial cells that make up brain capillaries [114,115]. Aβ aggregation was inhibited in the brain of animal models of AD using polymeric nanomicelles capable of releasing levels of 3D6 antibody fragments (3D6-Fab) [116]. Sophisticated immunotherapy techniques using single-chain anti-Aβ antibodies (scFv) significantly decreased Aβ load in an acute model of amyloidosis [117]. Therefore, there is a need for novel techniques to facilitate the delivery of such drugs into the brain and improve the efficacy of investigational new treatments.

##### Others

In addition to therapies that act on tau or β-amyloid pathology, there are other alternatives focused on reducing brain atrophy or symptomatology (Table 3). In this regard, several drugs are currently being studied for their ability to reduce the neuroinflammation that has been associated with AD, including TNF-α inhibitors, bacterial protease inhibitors, and selective tyrosine kinase inhibitors [118]. Other therapies that have also been the focus of attention in recent years are those aimed at reducing brain atrophy through the use of hepatocyte growth factors, or even through the administration of stem cells obtained from autologous adipose tissue [80]. Finally, another emerging pharmacological tool for the therapeutic approach to AD is the application of intranasal insulin (INI). Thus, recent studies based on the application of INI in AD patients have shown beneficial effects on the preservation of cognitive status, reducing inflammation and improving immune function, suggesting that treatment with INI may be a therapeutic alternative to preserve the proper development and maintenance of cognitive abilities [119,120].

### 2.3. Alternative Therapies

#### 2.3.1. Physical Activity

AD is a disease of multifactorial etiology where modifiable risk factors such as obesity, diabetes mellitus (DM), and sedentary lifestyle, among others, play an important role (Figure 3) [121]. In addition, it should be noted that AD can have a prolonged prodromal phase, which means that the development of the disease can begin as early as 25 years before the onset of clinical manifestations [122]. This presents an opportunity to establish interventions aimed at prevention and treatment of the disease at an early stage. As a consequence, the latest research on the treatment of AD has focused its work on the search for new non-pharmacological or preventive strategies to mitigate or halt the progression of the disease at an early stage, which is a great challenge given that current treatments are aimed at controlling the symptoms [123,124]. Among the different strategies being investigated, the adoption of a healthy lifestyle, including physical activity, a Mediterranean diet, and adequate sleep patterns, has shown promising results in terms of prevention [125,126].

To date, the scientific literature has shown that moderate physical activity (PA) not only helps to reduce the incidence of AD, but also leads to an improvement in functional and cognitive abilities, allowing these patients greater independence in carrying out the basic activities of daily living [127]. Thus, PA, and especially aerobic exercise, has a protective effect on cognitive function in the elderly and Alzheimer’s patients, as it is able to enhance the neuroplasticity of different brain structures [128]. Furthermore, several studies in humans and murine models have shown that PA represents a therapeutic alternative with the capacity to inhibit BACE-1 activity, enhance the clearance of Aβ-40 and Aβ-42, increasing the function of γ-secretase, and reducing tau hyperphosphorylation [129,130,131]. Similarly, it has been observed that exercise is able to improve insulin tolerance and glucose metabolism and utilization, which are impaired in patients with AD [132,133]. Finally, among the possible mechanisms of action that explain the benefits of PA in AD, it is worth highlighting the anti-inflammatory potential of exercise, capable of reducing levels of pro-inflammatory cytokines such as IL-6 or TNF alpha [134].

On the other hand, resistance exercise or strength training also show numerous benefits for AD. In this sense, several studies have shown that resistance training improves cognitive capacity and the development of basic activities of daily living, which could be explained by an increase in neuroprotective factor levels [135,136,137].

#### 2.3.2. Diet

In terms of diet, the Mediterranean diet offers a wide variety of nutrients with antioxidant and neuroprotective properties that preserve cognitive capacity and work on the main risk factors that increase the likelihood of AD [138]. In fact, the adoption of this type of dietary pattern has been found to be effective in reducing insulin resistance [139]. This finding is particularly important, as DM doubles the risk of dementia [140]. Among the different foods that comprise this diet, we can highlight those containing vitamin C (citrus fruits and some vegetables) and vitamin E (nuts, cereals and egg yolk), as they have been shown to reduce the accumulation of Aβ and improve cognitive capacity [138,139]. Along the same lines, the incorporation of foods rich in vitamin B in the diet has also shown positive effects, as they reduce brain atrophy in AD patients. This could be explained by the decreased action of homocysteine, a metabolite that causes increased oxidative activity, leading to increased neurodegeneration [139]. The consumption of fish rich in omega-3 fatty acids has also been shown to reduce the risk of cognitive decline [141]. It is also important to highlight the role of various compounds of plant origin, or polyphenols, which have antioxidant, anti-inflammatory, and neuroprotective properties. Among them, we can highlight some for their benefits in AD, such as curcumin from turmeric, epigallocatechin-3-gallate present in green tea, or resveratrol present in wine or blueberries; among others [142,143,144].

#### 2.3.3. Sleep Pattern

It has been shown that sleep disorders have a bidirectional relationship with the pathophysiology of AD, worsening the behavioral, cognitive, and memory alterations associated with this disease by increasing the β-amyloid load in response to an inflammatory process induced by these sleep disturbances [145]. In relation to this parameter, a field that has gained great interest in recent years focuses on the neuroprotective role of melatonin, as a reduction of this hormone has been observed in the early stages of AD [146]. Melatonin modulates the expression and function of secretase, thereby inhibiting the amyloidogenic processing of APP and Aβ production. Furthermore, it ameliorates Aβ-induced neurotoxicity by promoting Aβ clearance via glymphatic-lymphatic drainage, transport across the BBB, and degradation pathways [147,148,149]. In this regard, several clinical trials have studied the effect of exogenous melatonin administration in AD patients, with most of them showing an improvement in sleep quality and cognitive function [150,151,152,153]. Therefore, increased availability of melatonin could constitute a new therapeutic target [154]. In addition to this therapy, other alternatives to improve sleep patterns in patients with AD are being explored, such as bright light therapy and melatonin agonists such as ramelteon, agomelatine, and tasimelteon [155,156,157].

#### 2.3.4. Complementary Therapies

Finally, in addition to all the aforementioned strategies, the use of complementary medicine through aromatherapy and music therapy has proven to be effective in the management of behavioral symptoms and states of anxiety or depression, very common clinical manifestations in patients with AD [158]. The approach to AD also includes occupational therapy, where cognitive and behavioral activities are carried out to improve the quality of life of these patients and consequently, that of their caregivers as well [159].

## 3. Conclusions

The therapeutic approach to AD remains a challenge for researchers. Despite the advances that have been made in this field, we still do not have a treatment capable of halting the progress of the disease, as the therapies currently used have only achieved an improvement in the cognitive and behavioral deterioration of patients affected by this disease. In this sense, strategies based on the promotion of healthy lifestyles are indispensable in the prevention of the pathology, and these should be reinforced. Therefore, more research is needed to gain a more in-depth knowledge of AD, thus making it possible to identify new therapies that can contribute to its treatment.

## Figures and Tables

**Figure 1 pharmaceutics-14-01117-f001:**
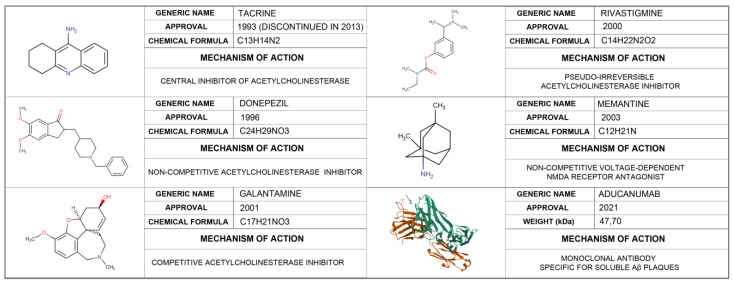
Drugs approved for the treatment of Alzheimer’s disease: mechanisms of action, weight, and molecular structure.

**Figure 2 pharmaceutics-14-01117-f002:**
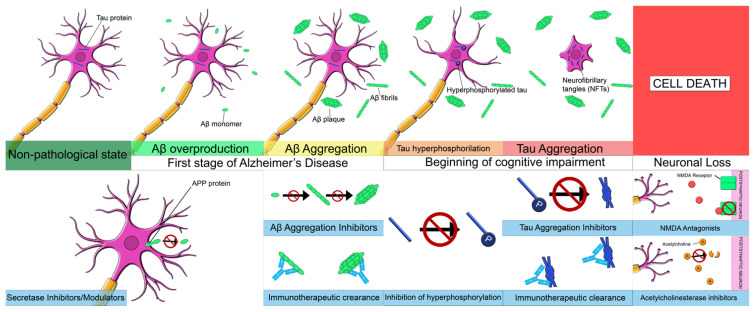
Molecular pathways involved in the pathology and in treatments under investigation for AD.

**Figure 3 pharmaceutics-14-01117-f003:**
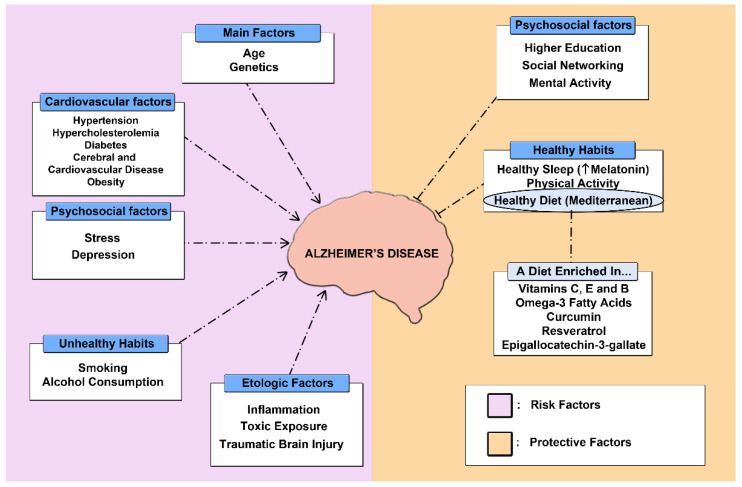
Diagram of protective and risk factors associated with Alzheimer’s disease.

**Table 1 pharmaceutics-14-01117-t001:** Pharmacological treatments under investigation related to Aβ pathology.

Pharmacological Treatments under Investigation			
	Mechanism of Action	Agent	
Aβ pathology	γ-secretase inhibitors	Semagacestat (LY-450139)	
		Avagacestat (BMS-708163)	
		Tarenflurbil	
	β-secretase inhibitors	Lanabecestat	
		Verubecestat	
		Atabecestat	
		Elenbecestat (E2609)	
		Umibecestat (CNP520)	
	α-secretase modulators	Etazolato (EHT0202)	
		APH-1105	
		ID1201	
	Aggregation inhibitors	Scyllo-inositol (ELND005)	
		Peptidomimetics (KLVFF, γ-AA)	
	Metal interfering drugs	Dyshomeotaisis (copper, iron or zinc) Deferiprona PBT2	
	Drugs that enhance Aβ clearance (immunotherapy)	Active immunotherapy	CAD106
			CNP520
			ABvac40
			GV1001
			ACC-001
			UB-311
			AF20513
		Passive immunotherapy	Crenezumab
			Gantenerumab
			LY3002813

**Table 2 pharmaceutics-14-01117-t002:** Pharmacological treatments under investigation related to tau pathology.

Pharmacological Treatments under Investigation			
	Mechanism of Action	Agent	
Tau pathology	Inhibitors of tau protein hyperphosphorylation	GSK3β inhibitors	Lithium Chloride
			Tideglusib
	Tau protein aggregation inhibitors	Methylene blue	
		TRx0237 (LMTM)	
	Drugs that promote the clearance of tau (immunotherapy)	Active immunotherapy	AADvac-1
			ACI-35
		Passive immunotherapy	C2N-8E12 (Tilayonemab)
			Bepranemab (UCB0107)
	Other anti-tau mAbs	BII076	
		JNJ-63733657	
		LY3303560	

**Table 3 pharmaceutics-14-01117-t003:** Other treatments under investigation.

Other Treatments under Investigation	
Agents	Nanomedicine strategies
	Intravenous immunoglobulin (IVIg)
	Plasma exchange via albumin
	TNF-α inhibitors
	Bacterial protease inhibitors
	Selective tyrosine kinase inhibitors
	Hepatocyte growth factors
	Stem cells
	Intranasal insulin

## Data Availability

Not applicable.

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
