# Peer review of "Therapeutic Approach to Alzheimer’s Disease: Current Treatments and New Perspectives"

_pharmaceutics, 2022, doi:10.3390/pharmaceutics14061117_

Round 1
Reviewer 1 Report
This manuscript summarizes the pathophysiology and therapeutic approach to Alzheimer ́s Disease (AD). The problem of this manuscript is that the contents is very similar to the author’s recent publication, “Current Understanding of the Physiopathology, Diagnosis and Therapeutic Approach to Alzheimer’s Disease. Biomedicines 2021, 9, 1910” (reference No. 5).
Further, I have following comments on this manuscript.
1. The authors should describe that European Medicines Agency rejected marketing authorization application for aducanumab.
2. The authors should make a table to summarize the pharmacological treatments under investigation.
Minor point
Page. 5, line 174, Tacrina=>Tacrine
Author Response
REVIEWER 1
This manuscript summarizes the pathophysiology and therapeutic approach to Alzheimer ́s Disease (AD). The problem of this manuscript is that the contents is very similar to the author’s recent publication, “Current Understanding of the Physiopathology, Diagnosis and Therapeutic Approach to Alzheimer’s Disease. Biomedicines 2021, 9, 1910” (reference No. 5).
Further, I have following comments on this manuscript.
Response: Thank you very much for your comments and suggestions which will contribute to improve the quality of our work.
The article you mention offers a global vision of the pathology of Alzheimer's disease, including a very detailed section on pathophysiology, another on diagnosis and a short section describing only some therapeutic options. The present work, however, deals in depth with the entire therapeutic arsenal available and under investigation to date, including other non-pharmacological therapeutic alternatives. Perhaps the introductory section is a little redundant, but it is intended to situate the reader before starting to read the results.
- The authors should describe that European Medicines Agency rejected marketing authorization application for aducanumab.
Response: Thank you very much for your suggestion. We have included a new sentence to clarify this issue.
Line 169-172: “However, the European Medicines Agency has withdrawn the marketing authorization for this product for the treatment of Alzheimer's disease due to interactions with the CHMP indicating that the data provided thus far would not be sufficient to support a positive opinion on the effectiveness of the product”
- The authors should make a table to summarize the pharmacological treatments under investigation.
Response: Thank you very much. Following your recommendation, we have made three new tables to summarize the pharmacological treatments under investigation and other strategies for AD.
Minor point
Page. 5, line 174, Tacrina=>Tacrine
Response: Sorry for the mistake. It has been modified

Reviewer 2 Report
This review is well written and of general interest.
I suggest some minor revisions to improve the presentation and scientific soundness:
1) please add some figures to better explain the molecular pathways involved in the pathology and in the therapeutic interventions
2) please add a table reporting potential drugs mentioned in the text other than those reported in figure 1
3) remove pullet poin and insert numbers for each sub-section
4) line 174 correct with tacrine
Author Response
REVIEWER 2
This review is well written and of general interest.
I suggest some minor revisions to improve the presentation and scientific soundness:
1) please add some figures to better explain the molecular pathways involved in the pathology and in the therapeutic interventions
Response: As suggested by the reviewer, we have included a new figure to explain the molecular pathways involved in the pathology and in the therapeutic interventions.
2) please add a table reporting potential drugs mentioned in the text other than those reported in figure 1
Response: Thank you very much for your comment. We have included three new tables to describe treatment under investigation and other alternative strategies for Alzheimer´s Disease
3) remove pullet poin and insert numbers for each sub-section
Response: Thank you very much for your suggestion. We have removed bullet points and insert instead numbers
4) line 174 correct with tacrine
Response: Sorry for the mistake. It has been modified

Reviewer 3 Report
Victoria García-Morales and coworkers excellently reviewed the topic entitled “Therapeutic approach to Alzheimer ́s Disease: current treatments and new perspectives” with a particular focus on the FDA-approved drugs, therapies under investigation for targeting Aβ pathology, and tau pathology, and alternative therapies designed to improving lifestyles. In summary, the authors provide an excellent summary that is worth getting published in the MDPI Pharmaceutics, however, I recommend a detailed revision addressing the following minor issues carefully:
1. The following English corrections should be corrected.
Page 2, lines 62 and 65: Authors used “SP are” SPs are
Page 3, line 131: N-methyl-d-aspartate receptor The locant should be italicized, the capital letter D should be used and the font of D should be smaller than usual. N-methyl-D-aspartate receptor
Page 5, line 194: Define AChE or add AchE in the brackets after Acetyl-cholinesterase in Page 5, line 173:
Page 2, line 85: blood-brain barrier was abbreviated as BBB. Later, on Page 5, line 180 and Page 12, line 523 blood-brain barrier was used.
2. I recommend the authors add another section as “nanomedicine strategies for Alzheimer’s treatment”. As the authors know that the current anti-AD drugs can only manage the clinical symptoms rather than preventing the progression or targeting the root cause of the disease. Nanomedicine strategies (Reference 1a,b,c) hold a tremendous potential for not only facilitating the targeted delivery to the specific part of the AD brain while sparing the healthy brain but also overcoming the hurdles associated with drug delivery such as BBB penetration using glucose-functionalized nanocarriers under fasting-induced glycemic control (Reference 2), off-targeting to peripheral organs (Reference 3), and treatment-related inflammation (Reference 4). Suggested references: Reference 1a: Nanocarrier mediated drug delivery as an impeccable therapeutic approach against Alzheimer's disease. J Control Release2022;343:528-550. doi: 10.1016/j.jconrel.2022.01.044. Reference 1b: Therapeutic strategies and nano-drug delivery applications in management of ageing Alzheimer's disease. Drug Deliv 2018;25(1):307-320. doi: 10.1080/10717544.2018.1428243. Reference 1c: Nanotechnologies for Alzheimer's disease: diagnosis, therapy, and safety issues. Nanomedicine2011;7(5):521-40. doi: 10.1016/j.nano.2011.03.008.
Reference 2: Dual-Sensitive Nanomicelles Enhancing Systemic Delivery of Therapeutically Active Antibodies Specifically into the Brain. ACS Nano 2020;14(6):6729-6742. DOI: 10.1021/acsnano.9b09991. Reference 3: Targeting nanoparticles to the brain by exploiting the blood-brain barrier impermeability to selectively label the brain endothelium. Proc Natl Acad Sci U S A 2020;117(32):19141-19150. DOI: 10.1073/pnas.2002016117. Reference 4: Improved Brain Expression of Anti-Amyloid β scFv by Complexation of mRNA Including a Secretion Sequence with PEG-based Block Catiomer. Curr Alzheimer Res 2017;14(3):295-302. DOI: 10.2174/1567205013666161108110031.

Author Response
REVIEWER 3
Victoria García-Morales and coworkers excellently reviewed the topic entitled “Therapeutic approach to Alzheimer ́s Disease: current treatments and new perspectives” with a particular focus on the FDA-approved drugs, therapies under investigation for targeting Aβ pathology, and tau pathology, and alternative therapies designed to improving lifestyles. In summary, the authors provide an excellent summary that is worth getting published in the MDPI Pharmaceutics, however, I recommend a detailed revision addressing the following minor issues carefully:
- The following English corrections should be corrected.
Response: Thank you very much for your comment. We have made the changes you suggest.
Page 2, lines 62 and 65: Authors used “SP are” à SPs are
Page 3, line 131: N-methyl-d-aspartate receptor à The locant should be italicized, the capital letter D should be used and the font of D should be smaller than usual. N-methyl-D-aspartate receptor
Page 5, line 194: Define AChE or add AchE in the brackets after Acetyl-cholinesterase in Page 5, line 173:
Page 2, line 85: blood-brain barrier was abbreviated as BBB. Later, on Page 5, line 180 and Page 12, line 523 blood-brain barrier was used.
- I recommend the authors add another section as “nanomedicine strategies for Alzheimer’s treatment”. As the authors know that the current anti-AD drugs can only manage the clinical symptoms rather than preventing the progression or targeting the root cause of the disease. Nanomedicine strategies (Reference 1a,b,c) hold a tremendous potential for not only facilitating the targeted delivery to the specific part of the AD brain while sparing the healthy brain but also overcoming the hurdles associated with drug delivery such as BBB penetration using glucose-functionalized nanocarriers under fasting-induced glycemic control (Reference 2), off-targeting to peripheral organs (Reference 3), and treatment-related inflammation (Reference 4).
Response: Thank you very much for your recommendation. We have included a new section about nanomedicine strategies for Alzheimer’s treatment.
Line 462-477: “Different nanobiomedicine treatments have been tested to facilitate targeted delivery to the specific part of the AD-affected brain while preserving the healthy brain, but also to overcome obstacles associated with drug delivery, such as BBB penetration. Nanocarriers have been made to increase the specificity of the drug to its target and are being tested to facilitate drug entry into the brain [112]. The blood-brain barrier has numerous im-pediments that limit drug entry, so nanotransporters, lipotransporters, carbon nanotubes and others have been tested to improve the efficacy and bioavailability of the drug in the central nervous system [113]. One solution to this problem is based on the design of nanoparticles with high affinity for the endothelial cells that make up brain capillaries [114-115]. Aβ aggregation was inhibited in the brain of animal models of AD using polymeric nanomicelles capable of releasing levels of 3D6 antibody fragments (3D6-Fab) [116]. Sophisticated immunotherapy techniques using single-chain anti-Aβ antibodies (scFv) significantly decreased Aβ load in an acute model of amyloidosis [117]. Therefore, there is a need for novel techniques to facilitate the delivery of such drugs into the brain and improve the efficacy of investigational new treatments”.

Round 2
Reviewer 1 Report
The authors revised adequately.